# A Novel *Bifidobacterium longum* ssp. *longum* Strain with Pleiotropic Effects

**DOI:** 10.3390/microorganisms12010174

**Published:** 2024-01-15

**Authors:** Merle Rätsep, Kalle Kilk, Mihkel Zilmer, Liina Kuus, Epp Songisepp

**Affiliations:** 1BioCC OÜ, Riia St. 181A, 50411 Tartu, Estonia; merle.ratsep@biocc.ee (M.R.);; 2Department of Biochemistry, Institute of Biomedicine and Translational Medicine, University of Tartu, Ravila St. 19, 50411 Tartu, Estonia

**Keywords:** *Bifidobacterium longum*, metabolomics, bioactive molecules, B vitamins, essential amino acid biosynthesis, milk, whey, postbiotic

## Abstract

Postbiotics are gaining increasing interest among the scientific community as well as at the level of food processing enterprises. The aim of this preliminary study was to characterise the metabolic diversity of a novel *Bifidobacterium longum* strain, BIOCC 1719, of human origin. The change after 24 h cultivation in three media was assessed using a metabolomic approach. Milk-based substrates favoured the activity of the strain, promoting the production of B vitamins, essential amino acids, bile acids, and fatty acids. Vitamins B1, B2, B6, B7, and B12 (with an average increase of 20–30%) were produced in both whole milk and whey; the increased production in the latter was as high as 100% for B7 and 744% for B12. The essential amino acids methionine and threonine were produced (>38%) in both milk and whey, and there was an increased production of leucine (>50%) in milk and lysine (126%) in whey. Increases in the content of docosahexaenoic acid (DHA) by 20%, deoxycholic acid in milk and whey (141% and 122%, respectively), and cholic acid (52%) in milk were recorded. During the preliminary characterisation of the metabolic diversity of the novel *B. longum* strain, BIOCC 1719, we identified the bioactive compounds produced by the strain during fermentation. This suggests its potential use as a postbiotic ingredient to enrich the human diet.

## 1. Introduction

The Western lifestyle and refined high-fat, high-sugar diets have altered the composition of host microbiota, leading to a deficiency of certain micronutrients and adverse effects on systemic human health. Various microbial metabolites have been well characterised, mostly in preclinical studies (in vitro, ex vivo, and animal models), regarding their health-promoting properties [1,2,3].

Bifidobacteria are commensals of the gastrointestinal tract (GIT), quickly colonising the infant intestine during the first weeks of human life and reaching the highest proportion in the colon during the first 12 months after birth. *Bifidobacterium longum* is one of the earliest and most successful colonisers of the GIT, and it is also widely distributed in the GIT of adults and the elderly, as intraspecific genomic diversity gives the species a competitive advantage and persistence in the host gut microbiome [4,5,6].

Although bifidobacteria represent only 3–6% of the adult gastrointestinal microbiota, they confer several positive health benefits to the host, contributing to gut barrier functions, gut, and immune homeostasis, and to protection against pathogens by competitive exclusion; during carbohydrate fermentation, they produce organic acids, which decrease the luminal pH and serve as cosubstrates for butyrate-producing colon bacteria through cross-feeding interactions [7,8].

Due to their positive health benefits, strains of different bifidobacteria species, including *B. longum*, have been most widely used as probiotics in a functional dose and a live form for supporting a healthy gut microbiota, but also for alleviating various conditions such as constipation, lactose intolerance, immunomodulation, oral infections, diarrhoeas of different ethology, and enteritis [9,10,11].

However, not all mechanisms or health benefits for a host are related to viable bacteria. Nonviable bacteria, their cell components, and bioactive metabolites also have impacts on human health.

According to the International Scientific Association for Probiotics and Prebiotics (ISAPP), a preparation of inanimate microorganisms and/or their components conferring a health benefit on the host is defined as a “postbiotic” [12]. According to the ISAPP, “inanimate” in postbiotics means that live microorganisms were present but have now been killed. Postbiotics can be differentiated by their composition (e.g., vitamins, organic acids) or by their physiological functions (immunomodulation, anti-inflammatory, hypocholesterolemic, antihypertensive, or antioxidative effects) [13].

The research and development of postbiotic products are of increasing interest for their application in an inanimate microbial culture and/or in using the bioactive metabolites in functional food products/supplements with technological advantages (e.g., longer product shelf life, no viability loss during manufacture and shelf life) or consumer-related advantages (e.g., lessened subjective side effects like bloating and flatulence, more favourable absorption of bioactive molecules) over products containing live microbes [13]. The spectrum of metabolites produced by the microbe in the specific growth medium/food matrix depends on the strain-specific properties of the microbe and the growth conditions and substrate characteristics.

Genomic diversity provides many strain-specific properties in *B. longum*, which, in turn, provides the opportunity to discover new strains that produce large numbers of bioactive metabolites with potential health effects.

Herein, we report a novel *Bifidobacterium longum* ssp. *longum* strain, BIOCC 1719 (DSM 34239), of healthy child origin. The aim of this preliminary study was to identify the metabolic diversity of the strain via the determination of a large number of compounds to verify possible pleiotropic health effects, keeping in mind the future possibilities of using the strain as a probiotic and/or postbiotic in the development of functional food products.

## 2. Materials and Methods

### 2.1. Origin of the Strain

The microorganism *Bifidobacterium longum* strain BIOCC 1719 (DSM 34239) (hereinafter BL1719) was isolated from a stool sample of a 2-month-old full-term breastfed Estonian child in 2020. The strain was identified by MALDI-TOF MS (Bruker MicroFlex Biotyper, Billerica, MA, USA) as *Bifidobacterium longum* ssp. *longum*. Identification was confirmed by PFGE analysis. The strain was characterised regarding safety and functional properties (patent pending).

### 2.2. Screening of BL1719 Metabolites in Different Growth Media

#### 2.2.1. Growth Media and Cultivation Conditions

A suspension from the 24 h old BL1719 culture was inoculated in a final inoculation dose of 5.5 log_10_ cfu/mL into 10 mL TPY broth (tryptone and papainic digest of soya, Condalab, Laboratorios Conda S.a., Madrid, Spain), autoclaved commercial cow milk (CM; fat content 2.5%, Valio Eesti AS Laeva Meierei, Estonia), and reconstituted demineralised sweet whey (RDSW, 10% of demineralised whey powder dissolved in distilled water, sterilised by autoclaving at 115 °C at 0.5 atm for 10 min). All versions were incubated anaerobically at 37 °C for 24 h. The viable count of BL1719 before and after cultivation in TPY, CM, and RDSW was measured by the serial dilution method on TPY agar.

The extracellular release of metabolites by BL1719 was examined from culture supernatants after the removal of microbial cells from TPY by centrifugation at 2000× *g* for 10 min at room temperature.

#### 2.2.2. Targeted Metabolic Profiling

Targeted metabolic profiling of TPY, CM, and RDSW was carried out with a Xevo TQ-XS mass spectrometer (Waters Corporation, Milford, CT, USA) coupled with ACQUITY ultraperformance liquid chromatography (Waters Corporation, Milford, CT, USA) using an MxP^®^ Quant 500 kit (Biocrates Life Sciences AG, Austria, https://biocrates.com/, accessed on 3 October 2023). This kit allows determination of over 630 different metabolites from one sample, including essential amino acids, amino acid-related compounds, fatty acids, and bile acids. The samples were prepared and analysed according to the manufacturer’s instructions. In brief, the samples were transferred onto filter papers preloaded with internal standards. Upon drying, all compounds containing amino groups were derivatised with phenylisothiocyanate and dried again. Finally, the metabolites were extracted with 5 mM ammonium acetate in methanol. The experiment was carried out without replicates.

#### 2.2.3. Short-Chain Fatty Acids (SCFAs) and B-Group Vitamin Analysis

The short-chain fatty acids (SCFAs) and B-group vitamins in TPY, CM, and RDSW were measured with Xevo TQ-XS mass spectrometry (Waters Corporation, Milford, CT, USA) coupled with ACQUITY ultraperformance liquid chromatography (Waters Corporation, Milford, CT, USA).

#### 2.2.4. Analysis of SCFAs

The SCFAs were derivatised with 2-nitrophenylhydrazine by treating a 50 μL sample with 50 μL internal standards (^2^H_4_acetic acid and ^2^H_11_hexanoic acid), 100 μL 200 mmol/L 2-nitrophenylhydrazine, and 20 μL 120 mmol/L 1-(3-dimethylaminopropyl)-3-ethylcarbodiimide. After 1 h of incubation, the mixture was centrifuged for 10 min at 21,000× *g*. The SCFAs were separated on an ACQUITY Premier BEH C18 Column with VanGuard FIT, 1.7 µm, 2.1 × 100 mm column using water and acetonitrile with 0.1% formic acid as eluents. 

#### 2.2.5. Analysis of B-Group Vitamins

The methodology for vitamins was as described by Yang and Rainville [14]. In brief, the vitamins were separated on an ACQUITY Premier BEH C18 Column with VanGuard FIT, 1.7 µm, 2.1 × 100 mm column using 20 mmol/L ammonium formate buffer in water and methanol with 0.1% formic acid.

### 2.3. Statistical Analysis

The effect of BL1719 on metabolites was calculated as the fold change in metabolite concentrations achieved after a 24 h fermentation period relative to the baseline concentrations of metabolites in unfermented sterile TPY, CM, and RDSW.

## 3. Results and Discussion

We characterised the metabolic capacity of the novel *Bifidobacterium* strain BL1719 via the determination of a large number of compounds, which provides an opportunity for more targeted analyses in the future. Herein, we present the de novo synthesis capacity of vitamin B regarding fermentation-induced changes in essential amino acids, bile acids, and fatty acids, i.e., compounds with the most documented effects on the human body. The 24 h cultivation time was chosen as it coincides with the end of the logarithmic growth phase of the strain and is short enough to be applied to the technological processes of the food industry. In this study, TPY broth served as a reference medium, as it was specially created for growing bifidobacteria. The results obtained in TPY provide useful information about the metabolic capacity of the *Bifidobacterium* strain under ideal conditions.

The viable count of BL1719 after 24 h cultivation was 8.4 log_10_ cfu/mL in TPY and CM and 7.9 log_10_ cfu/mL in RDSW.

### 3.1. Production of B-Group Vitamins

The cultivation of BL1719 for 24 h in TPY, an optimised laboratory growth medium for bifidobacteria, increased thiamine pyrophosphate (TPP, vitamin B1 coenzyme) content. The amount of TPP exceeded the initial concentration (0.007 mg/L) in the control sample (sterile uninoculated TPY) by 310% (Table 1). Also, a negligible (10%) increase in vitamin B12 was detected. Other vitamins initially present in TPY were utilised by the strain for its growth.

Compared to TPY, milk-based substrates promoted the production of B vitamins by the bifidobacteria strain. In turn, CM appeared to be the preferable substrate for BL1719, as a larger level of B vitamins was produced in CM than in RDSW. A similar increase (20–30%) in concentrations of thiamine (B1), riboflavin (B2), biotin (B7), and different forms of vitamin B6 (pyridoxine, pyridoxamine) compared to the control (i.e., sterile CM) was recorded. The results indicate that milk and milk products fermented with BL1719 could be a rich source of B vitamins. BL1719 affected forms of B6 in milk-based substrates differently, decreasing pyridoxal and increasing pyridoxine and pyridoxamine amounts in CM. In RDSW, an opposite trend was registered.

Though bifidobacteria are known to produce B-group vitamins, contrary reports exist regarding the ability of bifidobacteria to produce vitamins B2 and B7. According to Hosomi et al. [15], bifidobacteria lack the biosynthesis pathways for both.

Bifidobacteria are capable of breaking complex molecules and producing metabolic end products, including vitamins. Noda et al. [16] reported the effect of various carbon sources on increased biotin (B7) production by *B. bifidum*. Other reports argue that the presence of, e.g., riboflavin de novo synthesis genes appears to depend on the strain origin and can be found mainly in bifidobacteria of primate origin and, to a lesser extent, in strains of human origin [17]. Kwak and coworkers [18] identified complete riboflavin biosynthetic pathways in 4 *B. longum* strains among 44 different species of bifidobacteria strains of human infant origin. In the present study, B2 concentration increased in both milk-based substrates (i.e., CM and RDSW) fermented with BL1719. The amount of B2 increased by 24% in fermented CM from 23.567 mg/L in the control and by 42% from 28.191 mg/L in the control RDSW in fermented RDSW. Our results lead to the conclusion that BL1719, also of infant origin, is one of the rare human bifidobacterial strains that possesses the corresponding biosynthesis pathways and is capable of increasing the production of B2 and B7 in specific growth media. Alternatively, BL1719 can make the vitamins already present in the environment more available. For instance, B7 is usually covalently conjugated to proteins [19] and, for absorption, it needs to be released. Our detection method detects only noncovalently complexed or free compounds.

Various forms of B12, including cyanocobalamin, are synthesised by some bacterial species, including lactobacilli [20,21,22]. The biosynthesis of methylcobalamin by BL1719 was not detected in any of the tested growth media. A negligible increase in adenosylcobalamin was registered only in CM. On the other hand, in RDSW, a surprisingly high increased production (744%) of cyanocobalamin (B12) was detected (Table 1). Cyanocobalamin is the most stable (including during thermal processing) and therefore the most widely used form of B12 in food supplements and in fortified functional foods for the alleviation of B12 deficiency [23]. After ingestion, to obtain a bioactive form of B12, all the cobalamin forms (including cyanocobalamin) are adsorbed in the host intestine, internalised by the epithelial cells, and, thereafter switched to an intracellular pathway and then converted to the functional coenzymes methylcobalamin or adenosylcobalamin. B12 adsorption in the intestine is limited to 1.5 µg per meal [24].

The metabolic activity of BL1719 in RDSW may be partially contributed to by the intracellular nutrients and cell particles released from the starter cultures killed during the technological processing of cheese whey into demineralised whey powder and the autoclaving during RDSW preparation. The effect of whey proteins on the ability of BL1719 to biosynthesise vitamin B12 needs further investigation. Our results indicate that RDSW fermented by BL1719 can be a good substrate for the development of a B12-enriched additive for various functional foods.

Vitamin B3 was utilised by BL1719 for growth in all tested growth media.

According to the Nordic Nutrition Recommendation [25], the Recommended Daily Values (RDVs) for adults of both sexes for B2 is 1.6 mg/d and for B7 and B12 is 0.04 mg/d. The fermentation of CM and RDSW with strain BL1719 managed to enrich both substrates with B2, B7, and B12, with the 100 mL intake exceeding the RDV in CM by 182.1% for B2, by 1105% for B7, and by 25.0% for B12. In RDSW, the respective values were by 250.4% for B2, by 124.5% for B7, and by 162.5% for B12.

### 3.2. Production of Essential Amino Acids and Amino-Acid-Related Compounds

Although the biosynthetic and degradation pathways for vitamins, amino acids, cofactors, and noncarbohydrate substances have been reported in bifidobacteria, the amino acid utilisation and the metabolism of bifidobacteria are not fully understood [26,27].

In the present study, after 24 h of BL1719 cultivation, a 45% increase in amino acid taurine was detected in TPY compared to the initial amount of 1.4 µmol/mL taurine in the control. In CM, the increase remained smaller (i.e., 10%), and in RDSW, BL1719 used up all the taurine initially available. The high increase in taurine in TPY in comparison with the other substrates makes sense, as taurine is derived from cysteine, which is present in TPY broth (0.5 g/L). Due to the lack of genes encoding the sulphur assimilation pathway, most bifidobacteria are auxotrophic for cysteine because they cannot synthesise cysteine intracellularly and need a cysteine-supplemented growth medium. BL1719 prefers cysteine-supplemented growth media. The strain is also able to grow (though poorly) in environments not fortified with cysteine.

Regarding essential amino acids, the largest fermentation-induced change was detected in six amino acids in the tested growth media (Table 2). BL1719 increased the amount of the essential amino acid methionine in all the tested media, with the increase being higher in milk-based substrates, where the change exceeded 50% in both CM and RDSW (Table 2). In CM, methionine increased by 58% from 3.97 µmol/L in the control and by 56% from 2.25 µmol/L in the control. Methionine is essential in several physiological and biochemical processes in the human body through being a key donor of methyl groups and possessing a reduced sulphur atom, which is essential for redox balance [28]. Strains of *B. longum* ssp. *longum* species have been reported to possess genes for several amino acid biosynthesis pathways, including methionine [26]. On the other hand, there are indications that some bifidobacterial strains of different species are able to synthesise cysteine and methionine over homocysteine in the S-Adenosyl-L-Methionine (SAM) cycle [27]. We detected a 77% increase in homocysteine in RDSW from 0.347 µmol/L to 0.614 µmol/L in parallel with the increase in methionine due to the fermentation with BL1719. Whether or not BL1719 possesses a reverse trans-sulphuration pathway remains to be clarified.

After BL1719 cultivation, phenylalanine and tryptophan increased in TPY (60% and 31%, respectively). In milk-based substrates, the strain utilised both these amino acids for growth. In addition to being an essential amino acid for building proteins, phenylalanine is a starting point for the synthesis of catecholamines (dopamine, norepinephrine, and epinephrine) as well as pigments (melanins). Tryptophan is a precursor for serotonin and kynurenine. The former is an important regulator of bowel health and microbiota–host communication [29]. Kynurenine is a regulator of the immune system and, together with the rest of the tryptophan catabolic metabolites, is a signalling molecule in microbiota–host and gut–brain communication [30].

After fermentation, an increase in the concentration of the essential amino acid leucine of 46% was detected in TPY and in CM, where the amino acid increased by 52% compared to the initial 11.53 µmol/L in the control. Leucine is known to affect hormonal activity by regulating glucose homeostasis and insulin resistance, among other effects, although the mechanisms still need to be understood [31]. Leucine is also a well-known and widely used food supplement for muscle buildup and energy production. During the fermentation of RDSW, leucine was utilised for growth by BL1719.

Another essential amino acid increasingly produced by BL1719 in all tested media was threonine. In TPY, the change (+21%) remained low, and both milk-based substrates appeared to be more favourable (Table 2). Namely, the increase was over 38% in CM and over 65% in fermented RDSW in comparison with the respective control. In comparison with milk, whey proteins are known to be richer in lysine, threonine, and leucine [32].

Threonine prevents fatty buildup in the liver and is one of the aminodetoxifiers. Li et al. [33] found, in a community-based case-control study, that the consumption of lysine-, threonine-, and valine-rich foods helps to reduce nonalcoholic fatty liver disease risk in the elderly population.

Among the essential amino acids, the highest overproduction (i.e., 126% in RDSW) after fermentation by BL1719 was detected in the case of lysine (Table 2). In the human body, lysine is one of the precursors for l-carnitine.

Though valine and histidine biosynthesis genes have been reported by Milani et al. [26] to be present in species *B. longum* ssp. *longum*; BL1719 used both amino acids for growth.

### 3.3. Production of Fatty Acids

Regarding SCFA production, cultivating BL1719 for 24 h increased the amount of acetic acid (by 442% from the initial 18 µmol/L) and caproic acid (by 34% from the initial 0.004 µmol/L) only in TPY; in milk-based substrates, no increase was recorded (Table 3). On the other hand, some increases in caprylic and pimelic acids were detected only in CM (by 42% from the initial 0.031 µmol/L and by 26% from the initial 0.001 µmol/L, respectively).

Lactic acid was the main organic acid produced by BL1719 in the growth medium. The highest increase (1408%) was detected in lactic acid in TPY from the initial 366,718 µmol/L, with a 383% increase in CM from 348.5 µmol/L and a 129% increase in RDSW from 818.5 µmol/L (Table 3). Though, upon their discovery in 1924, bifidobacteria were initially classified as lactic acid bacteria, since lactic acid is one of the main fermentation end products, they have been reclassified. Bifidobacteria are phylogenetically distinct, and their enzymatic profile differs from that of lactic acid bacteria. In bifidobacteria, the unique carbohydrate metabolism, called the bifidus pathway (or the fructose-6-phosphate shunt), is a major pathway of carbohydrate metabolism [34]. The final fermentation products include mainly acetic acid but also lactic acid, formic acid, ethanol, succinic acid, and, in some species, also butyric acid, and propionic acid. Although carbohydrate metabolic abilities vary between different species of bifidobacteria, all bifidobacteria of human origin can use fructose, glucose, galactose, and lactose for life [34,35]. Though whey is relatively rich in lactose, it is low in other nutrients. In comparison with whey, whole cow milk has a better nutritional composition as it contains more proteins, fats, carbohydrates, vitamins, and trace elements. TPY is a medium specifically designed for the cultivation of bifidobacteria and thus supports the metabolism of the species of this genus best in comparison with the milk-based substrates used in this study. This could be the main reason why a 14.08-fold increase in lactic acid in TPY was detected and the respective results remained lower in CM and RDSW. Succinic acid was also produced by BL1719 in all growth media during growth, increasing the initial amount of 503.5 µmol/mL by 282% in TPY; in milk-based substrates, the results were lower by +38% in CM from 126.5 µmol/mL and by +64% RDSW from 30.5 µmol/L.

Propionic and butyric acids did not increase or had negligible increases in all the substrates.

BL1719 also increased the amounts of DHA (docosahexaenoic acid, a very long omega-3 polyunsaturated fatty acid with cardiovascular health-supporting properties) in comparison with the respective controls in all tested media, where the respective increase in TPY was 30% to 0.773 µmol/L from 0.5945 µmol/L in the control. In CM and RDSW, the increase was 20% from the baselines in the respective controls (1.47 µmol/L and 0.956 µmol/L). In the future, it will be important to investigate whether a longer fermentation time helps to increase the concentration of DHA in milk-based substrates.

### 3.4. Production of Bile Acids

In TPY, BL1719 utilised bile acids for growth. In CM, an increase in cholic acid (CA, the primary bile acid) of 52% from 0.123 µmol/L and an increase in deoxycholic acid (DCA, the secondary unconjugated bile acid) of 141% from 0.123 µmol/L in the control were detected. DCA also increased by 122% to 0.019 µmol/L in fermented RDSW from the initial 0.0185 µmol/L in the control. About 95% of bile acids (including CA) are adsorbed in the terminal ileum and are transported back to the liver and to be recycled. About 5% of bile acids pass into the colon and are modified by the colon microbiota to increase and enrich the diversity of the composition of the bile acid pool. Deconjugated primary bile acids are enzymatically transformed to secondary bile acids (e.g., DCA), carried out by bile salt hydrolase activity by representatives of the gut microbiota [36,37]. DCA is the end product of an enzymatic process carried out by salt hydrolase activity. Our results imply the possible bile salt hydrolase activity of the strain. The presence of corresponding genes has been reported in representatives of the same species [26]. The specific properties of B1719 bile salt should be identified in future studies.

## 4. Conclusions

In the initial characterisation of the strain-specific metabolic diversity of the novel *Bifidobacterium longum* strain BIOCC 1719, we determined various bioactive compounds that the strain simultaneously produced in the growth medium during fermentation. In milk-based substrates (whole milk and whey), increases in vitamins B2, B7, and B12; several essential amino acids (leucine, lysine, methionine, and threonine); and lactic and succinic acids were registered. This study justifies the use of the properties of this strain in the future to develop postbiotic ingredients to enrich human nutrition. The health effects on the host must also be tested to confirm the pleiotropic effects of the viable or inactivated strain and its bioactive substances on the gastrointestinal microbiota and blood parameters.

## 5. Patents

Authors are inventors of International Patent Application PCT/IB2023/063030.

## Figures and Tables

**Table 1 microorganisms-12-00174-t001:** Concentrations of B-group vitamins (mg/L) in TPY broth, cow milk, and reconstituted demineralised sweet whey and changes (%) after 24 h of fermentation with BL1719.

	TPY *		CM *		RDSW *	
	Control **	Change ***	Control	Change	Control	Change
Thiamine (B1)	0.216	−87	0.167	+22	0.101	−4
Thiamine pyrophosphate (TPP, B1 coenzyme)	0.007	+310	0.00005	−24	0.0000	nc ^#^
Riboflavin (B2)	6.313	−16	23.567	+24	28.191	+42
Pyridoxine (B6)	0.230	−23	0.002	+23	0.004	−29
Pyridoxal (B6)	0.007	−50	0.647	−2.2	0.885	+25
Pyridoxamine (B6)	0.043	−23	0.084	+23	0.032	−18
Biotin (B7)	1.578	−7	3.437	+29	0.498	+100
Cyanocobalamin (B12)	135.098	+10	0.029	−64	0.008	+744

* TPY—tryptic soy yeast extract broth; CM—cow milk; RDSW—reconstituted demineralised sweet whey; ** Control—respectively, uncultivated sterile TPY, CM, or RDSW; *** Change—in comparison with the control; ^#^ nc—incalculable.

**Table 2 microorganisms-12-00174-t002:** Absolute concentrations (µmol/L) of essential amino acids in TPY broth, milk, and reconstituted demineralised sweet whey and changes (%) after 24 h of fermentation.

Amino Acid	TPY *		CM *		RDSW *	
	Control **	Change ***	Control	Change	Control	Change
Leucine	3114.00	+46	11.53	+52	11.32	−25
Lysine	0.00	0	65.00	−39	21.22	+126
Methionine	844.50	+44	3.97	+58	2.25	+56
Phenylalanine	5512.50	+60	9.52	−0.32	5.99	−0.33
Threonine	1953.00	+21	12.30	+38	9.92	+65
Tryptophan	552.500	+31	5.85	−23	2.72	−27
Histidine	453.50	−1.8	11.30	−20	5.08	+6
Valine	10,862.00	−3	18.90	+13	17.05	−4.4

* TPY—tryptic soy yeast extract broth; CM—cow milk; RDSW—reconstituted demineralised sweet whey; ** Control—respectively, uncultivated sterile TPY, CM, or RDSW; *** Change—in comparison with the control.

**Table 3 microorganisms-12-00174-t003:** Absolute concentrations (µmol/L) of SCFAs and lactic and succinic acids in TPY broth, milk, and reconstituted demineralised sweet whey and changes (%) after 24 h of fermentation.

	TPY *		CM *		RDSW *	
	Control **	Change ***	Control	Change	Control	Change
Acetic acid (C2)	17.569	+441.7	15.328	−2.5	0.030	−0.5
Propionic acid (C3)	0.026	−12	0.029	−12	0.075	−22.5
Butyric acid (C4)	0.089	−8	0.0628	−5.3	0.0751	−82
Valeric acid (C5)	0.001	+23	0.001	+11	0.001	−13
Caproic acid (C6)	0.004	+34	0.017	−0.8	0.013	−83
Pimelic acid (C7)	0.001	+2.0	0.001	+26	0.001	−59
Caprylic acid (C8)	0.008	−25	0.031	+42	0.029	−67
Lactic acid	3667.2	+1408	348.5	+383	818.5	+129
Succinic acid	503.5	+282	126.5	+38	30.6	+64

* TPY—tryptic soy yeast extract broth; CM—cow milk; RDSW—reconstituted demineralised sweet whey; ** Control—respectively, uncultivated sterile TPY, CM, or RDSW; *** Change—in comparison with the control.

## Data Availability

Data are contained within the article.

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
