# Peer review of "A Novel Bifidobacterium longum ssp. longum Strain with Pleiotropic Effects"

_microorganisms, 2024, doi:10.3390/microorganisms12010174_

Round 1
Reviewer 1 Report
Comments and Suggestions for Authors
This article introduces an interesting experiment. The author first isolated Bifidobacterium longum strain BIOCC 1719 (DSM 34239) from a 2-month-old full-term breastfed child. This strain was then cultured in tryptic soy yeast extract broth (TPY), autoclaved commercial cow milk (CM), and reconstituted demin eralised sweet whey (RDSW). The study investigated the metabolic products of BL1719 in different growth media. Overall, the results of the experiment showed that Milk-based substrates favoured the activity of the strain, promoting the production of B vitamins, essential amino acids, bile acids and fatty acids. This suggests that BL1719 has potential for developing functional foods.
The experimental design of this article is reasonable, the data processing is accurate, the logic of the paper is clear, and the expression is standardized. Overall, it is a well-written paper.
Here are some comments:
1. Line 87. It should be noted that there is a problem of redundant expression and inconsistent use of parentheses here, which states 'tryptic soy yeast extract broth (TPY) broth, Condalab, Spain)'. Please make the necessary revisions.
2. The section of "3. Results and Discussion" should be divided into multiple subheadings, like “3.1 ... ...; 3.2 ... ...; 3.3 ... ...” to present the research findings in a more organized manner.
3. Please explain why the author uses tryptic soy yeast extract broth (TPY) instead of regular broth.
Author Response
Response to Reviewer 1 Comments
|
||||
1. Summary |
|
|
||
Thank you very much for taking the time to review this manuscript. Please find the detailed responses below in the point-by-point response letter and the corresponding corrections highlighted in the re-submitted file.
|
||||
2. Questions for General Evaluation |
Reviewer’s Evaluation |
Response and Revisions |
||
Does the introduction provide sufficient background and include all relevant references? |
Yes |
|
||
Are all the cited references relevant to the research? |
Yes |
|
||
Is the research design appropriate? |
Yes |
|
||
Are the methods adequately described? |
Yes |
|
||
Are the results clearly presented? |
Yes |
|
||
Are the conclusions supported by the results?
|
Yes |
|
||
3. Point-by-point response to Comments and Suggestions for Authors |
||||
Comments 1: Line 87. It should be noted that there is a problem of redundant expression and inconsistent use of parentheses here, which states 'tryptic soy yeast extract broth (TPY) broth, Condalab, Spain)'. Please make the necessary revisions. |
||||
Response 1: Thank you for pointing this out. The mistake has been corrected. The correction in revised manuscript is on page 2, lines 87-88:
|
||||
Comments 2: The section of "3. Results and Discussion" should be divided into multiple subheadings, like “3.1 ... ...; 3.2 ... ...; 3.3 ... ...” to present the research findings in a more organized manner Response 2: Thank you for pointing this out. We have accordingly modified the revised manuscript. Following subheadings added to “Results and Discussion”. 3.1. Production of B-group vitamins (page 3, line 143) 3.2. Production of essential amino acids and amino acid-related compounds (page 5, line 213) 3.3. Production of fatty acids (page 8, line 280) 3.4. Production of bile acids (page 8, line 317)
Comments 3: Please explain why the author uses tryptic soy yeast extract broth (TPY) instead of regular broth. Response 3: Thank you for the question. First, sorry for the misspelling: the correct name of the TPY broth is tryptone and papainic digest of soy. The mistake has been corrected. The correction in the revised manuscript is on page 2, line 88: A suspension from the 24 h old BL1719 culture was inoculated in final inoculation dose 5.5 log10 cfu /mL into 10 mL TPY broth (tryptone and papainic digest of soy, Condalab, Spain), autoclaved commercial cow milk (CM; fat content 2.5%, Valio) and reconstituted demineralised sweet whey (RDSW, 10% of demineralised whey powder dissolved in distilled water, sterilised by autoclaving at 115 °C. 0.5 atm for 10 min).
The TPY broth was selected, as it is a medium specifically designed for cultivation of Bifidobacteria and provides nutrients essential for bifidobacterial growth thus supporting the metabolism of species this genus better in comparison with other culture media. In present study, the TPY broth served as a reference medium in comparison with milk-derived substrates. The explanation has been added to the revised manuscript (page 3, lines 136-139).
|
||||
|
||||
Reviewer 2 Report
Comments and Suggestions for Authors
A novel Bifidobacterium longum ssp. longum strain with pleiotropic effects.
By Merle Rätsep, Kalle Kilk, Mihkel Zilmer, Liina Kuus, and Epp Songisepp
Manuscript Number:
The review
Summary Statement:
Bifidobacterium longum strain BIOCC 1719, which originates from humans, has been reported to be a source of essential B-group vitamins, including B12, amino-bile, and fatty acids.
When cultured on milk and whey-based media, the strain showed an increase in docosahexaenoic, cholic, and deoxycholic acids. The diversity of bioactive compounds allows for the use of the strain culture fluid as a postbiotic ingredient in functional foods.
Major Strengths
Bifidobacterium longum strain BIOCC 1719 was isolated from a breastfed 2-month-old child. The strain was cultured in cow milk and milk-derived media, as well as tryptic soy/yeast extract. The main metabolites, including several amino acids, SCFA, and B-group vitamins, were profiled using LC-MS. A significant increase in B12 cyanocobalamin was detected in the whey-based medium. The study analyzed the profile of essential amino acids, such as lysine, methionine, phenylalanine, and threonine, as well as short-chain fatty acids, based on the culturing media.
The manuscript is well-structured.
Areas of Improvement
Minor Points:
1. Materials and methods: Amino acid profiling protocol should be submitted along with SCFA and B-vitamin LC-MS protocol.
2. Sample preparation procedure. Simple centrifugation of Bifidobacterium culture broth removes a clear majority of cells, while proteins and polysaccharides, which could alter the results of chromatographic profiling, remain more or less intact in the culture fluid. In addition, these substances could easily clog the 1.7 um LC column with high acetonitrile/methanol phase and alter the results.
3. Please comment on why the metabolic profiling experiments were not replicated (Line 102).
4. Results and discussions: Lines 80-83 The strain was identified using Bruker Biotyper and its internal database. Additionally, the origin was confirmed through PFGE analysis. The analysis results and confirmation of the strain's novelty are important for readers to know. A description or reference of the strain’s novelty should be provided.
5. Cell concentrations after 24 hours of cultivation in different media are presented in lines 131-132. It would be valuable to include information on the approximate concentration of wet biomass in either grams per liter or as a percentage of the culture broth.
6. Table 2 shows that the B12 concentration in TPY media increased by 10% from 135 mg/L to approximately 149 mg/L. This is a significant finding. Please provide your comments.
7. Table 3 and lines 266-267 report the lactic acid concentration in TPY media. The initial concentration was 0.367M, and after 24 hours, it increased 14.08-fold to 5.17M or 465 g/L. The production rate of lactic acid was 18 g/L/h, which is a noteworthy result. Please provide your comments.
8. Compare the data obtained to the Recommended Daily Values (RDV) for vitamins and essential amino acids, if applicable, and consider the appropriate amount of culture fluid for food additives needed.
9. The standard deviation (SD) values for the analysis results would be greatly appreciated.
Conclusion:
The manuscript is suitable for publication in Microorganisms with minor corrections. However, several topics require deep clarification.

Minor editing of English and proofreading required
Author Response
Response to Reviewer 2 Comments
|
|||
1. Summary |
|
|
|
Thank you very much for taking the time to review this manuscript and for your good suggestions. Please find the detailed responses below in the point-by-point response letter and the corresponding revisions/corrections highlighted in the re-submitted file.
|
|||
2. Questions for General Evaluation |
Reviewer’s Evaluation |
Response and Revisions |
|
Does the introduction provide sufficient background and include all relevant references? |
Can be improved |
|
|
Are all the cited references relevant to the research? |
Yes |
|
|
Is the research design appropriate? |
Must be improved |
response in the point-by-point response letter. |
|
Are the methods adequately described? |
Must be improved |
Response is in the point-by-point response letter. |
|
Are the results clearly presented? |
Must be improved |
response in the point-by-point response letter. |
|
Are the conclusions supported by the results? |
Must be improved
|
improved |
|
3. Point-by-point response to Comments and Suggestions for Authors |
|||
Comment 1: Materials and methods: Amino acid profiling protocol should be submitted along with SCFA and B-vitamin LC-MS protocol. |
|||
Response 1: Thank you for pointing this out. Amino acid profiling was carried out by targeted metabolic profiling using the MxP® Quant 500. The corresponding addition has been added to Materials and Methods, 2.2.2. Targeted metabolic profiling (page 3, Lines 101-107). |
|||
Comment 2: Sample preparation procedure. Simple centrifugation of Bifidobacterium culture broth removes a clear majority of cells, while proteins and polysaccharides, which could alter the results of chromatographic profiling, remain more or less intact in the culture fluid. In addition, these substances could easily clog the 1.7 um LC column with high acetonitrile/methanol phase and alter the results. |
|||
Response 2: The samples were processed according to our biochemist’ instructions, who has been involved in several microbial metabolite analyses by LC, including from milk-based substrates. This question is answered by the added description of methodology (see answer to comment 1). Sample preparation involves several drying steps, and the final sample is obtained by centrifugation of extraction solution through the filter paper on which the derivatization is done. The biopolymers remain in the filter paper and do not interfere with the analysis.
Comment 3: Please comment on why the metabolic profiling experiments were not replicated (Line 102). Response 3: Thank you for pointing this out. It was a preliminary study (as also mentioned Line 72) to get possibly broad-based information about the metabolites of the novel Bifidobacterium strain. The experiment was carried out without replicates for the following reasons: the MxP® Quant 500 kit allows screening over 630 metabolites from one sample, which results in a very large amount of raw data. Since the strain was tested in three growth media, the amount of data tripled, and it took an unexpected amount of time to process. Additionally, the kit MxP® Quant 500 quite expensive and it was the first time for us use to it. We will repeat the experiment, planning to organize a more targeted analysis based on the results of the present Study.
Comment 4. Results and discussions: Lines 80-83 The strain was identified using Bruker Biotyper and its internal database. Additionally, the origin was confirmed through PFGE analysis. The analysis results and confirmation of the strain's novelty are important for readers to know. A description or reference of the strain’s novelty should be provided. Response 4 Thank you for pointing this out. As mentioned in the Materials and Methods, the strain Bifidobacterium longum strain BIOCC 1719 (DSM 34239) was isolated in 2020. Since isolation, the properties of this strain have been tested in an interdisciplinary manner. It is a proprietary strain of BioCC OÜ. The strain is patent pending, international patent application PCT/IB2023/063030 was filed on 20.12.2023. Present study being the first public mentioning of the strain. Other manuscripts are in compilation stage.
Comment 5. Cell concentrations after 24 hours of cultivation in different media are presented in lines 131-132. It would be valuable to include information on the approximate concentration of wet biomass in either grams per liter or as a percentage of the culture broth. Response 5: Thank you for pointing this out. During the experiment, the wet biomass was not measured. Considering the experimental design, we considered the determination of the number of viable cells using the serial dilution method more relevant, as the strain was cultivated in 10 mL of growth medium (the volume of the growth medium has been added into Material and Methods, page 2, Line 88), thus the wet biomass would have been relatively small, and the weighing results would have been inaccurate, as due to surface tension it is not possible to remove all the supernatant from a 10 ml tube.
Comment 6. Table 2 shows that the B12 concentration in TPY media increased by 10% from 135 mg/L to approximately 149 mg/L. This is a significant finding. Please provide your comments. Response 6: TPY broth is a medium specifically designed for cultivation of bifidobacteria. It comprises a rich array of nutrients including yeast extract (2.5g/L). Therefore, the initial amount of B12 was also high, in comparison with other vitamins or other growth media. However, the 10% increase of B12 in TPY remained much lower than the change in whey, where the increase was over 700%. Although the final concentration of B12 in TPY medium high, TPY is still a laboratory medium. The result provides useful information about the metabolic capacity of the Bifidobacterium strain under ideal conditions. On the other hand, whey is a food-grade substrate. The increased production of B12 by the Bifidobacterium strain in whey is therefore more significant, as the use of fermented whey offers an economically and environmentally friendly way to use the properties of the strain in supporting human health.
Comment 7. Table 3 and lines 266-267 report the lactic acid concentration in TPY media. The initial concentration was 0.367M, and after 24 hours, it increased 14.08-fold to 5.17M or 465 g/L. The production rate of lactic acid was 18 g/L/h, which is a noteworthy result. Please provide your comments. Response 7: Thank you for pointing that out. Bifidobacteria are lactic acid bacteria. Sugars (incl. lactose) are fermented into lactic acid, which is excreted from the bacterial cell. Though whey is relatively rich in lactose, it is low in other nutrients. In comparison with whey, cow milk has a better nutritional composition, as it contains protein, fat, carbohydrates, vitamins, and trace elements. However, TPY is a medium specifically designed for cultivation of Bifidobacteria and supports the metabolism of the species this genus best in comparison with milk-based substrates used in this study. This could be the main reason why the highest increase in TPY was detected and the results remained lower, in CM and RDSW. The corresponding reasoning has also included into the discussion (page 7, lines 295-303)
Co Comment 8. Compare the data obtained to the Recommended Daily Values (RDV) for vitamins and essential amino acids, if applicable, and consider the appropriate amount of culture fluid for food additives needed. Response 8: Thank you, we agree with this comment. To emphasize this point, we have updated the text in the manuscript (page 5, lines 206-211): According to Nordic Nutrition Recommendation (2023), the Recommended Daily Values (RDV) for adults of both sexes for B2 is 1.6 mg/d, for B7 and B12 is 0.04 mg/d. The fermentation of CM and RDSW with strain BL1719 managed to enrich both substrates with B2, B7 and B12, 100 mL intake exceeding the RDV in CM by 182.1% for B2, by 1,105% for B7 and by 25.0% for B12. In RDSW the respective values were by 250.4% for B2, by 124.5% for B7 and by 162.5% for B12. No RDV for amino acids intake are suggested according to the Nordic Nutrition Recommendation (2023). For protein intake an adequate requirement is 0.66 g/kg/d and recommended intake for adults is 0.83 g/kg/d, essential amino acids should cover 10-20% of amino acids pool, i.e., between 0.06 to 0.12 g/kg per day for adult.
Comment 9. The standard deviation (SD) values for the analysis results would be greatly appreciated. Response 9: As it was a preliminary study to get possibly broad-based information about the novel Bifidobacterium strain, the experiment was carried out without replicates. The experiments will be repeated in a more targeted manner.
|
|||
4. Response to Comments on the Quality of English Language |
|||
Point 1: Minor editing of English language is required |
|||
Response 1: The manuscript has been checked by native speaker. All changes are highlighted in revised manuscript.
|
|||
5. Additional clarifications |
|||
The manuscript has undergone language revision by a native English speaker.
